# Macrofluidic Coaxial Flow Platforms to Produce Tunable Magnetite Nanoparticles: A Study of the Effect of Reaction Conditions and Biomineralisation Protein Mms6

**DOI:** 10.3390/nano9121729

**Published:** 2019-12-04

**Authors:** Laura Norfolk, Andrea E Rawlings, Jonathan P Bramble, Katy Ward, Noel Francis, Rachel Waller, Ashley Bailey, Sarah S. Staniland

**Affiliations:** 1Department of Chemistry, University of Sheffield, Brook Hill, Sheffield S3 7HF, UK; lnorfolk1@sheffield.ac.uk (L.N.); a.rawlings@sheffield.ac.uk (A.E.R.); jbramble82@gmail.com (J.P.B.); klward2@sheffield.ac.uk (K.W.); nfrancis1@sheffield.ac.uk (N.F.); rwaller2@sheffield.ac.uk (R.W.); 2School of Physics and Astronomy, University of Leeds, Leeds LS2 9JT, UK; bailey272@hotmail.co.uk

**Keywords:** fluidic, magnetite, magnetic nanoparticle, flow synthesis, Mms6

## Abstract

Magnetite nanoparticles’ applicability is growing extensively. However, simple, environmentally-friendly, tunable synthesis of monodispersed iron-oxide nanoparticles is challenging. Continuous flow microfluidic synthesis is promising; however, the microscale results in small yields and clogging. Here we present two simple macrofluidics devices (cast and machined) for precision magnetite nanoparticle synthesis utilizing formation at the interface by diffusion between two laminar flows, removing aforementioned issues. Ferric to total iron was varied between 0.2 (20:80 Fe^3+^:Fe^2+^) and 0.7 (70:30 Fe^3+^:Fe^2+^). X-ray diffraction shows magnetite in fractions from 0.2–0.6, with iron-oxide impurities in 0.7, 0.2 and 0.3 samples and magnetic susceptibility increases with increasing ferric content to 0.6, in agreement with each other and batch synthesis. Remarkably, size is tuned (between 20.5 nm to 6.5 nm) simply by increasing ferric ions ratio. Previous research shows biomineralisation protein Mms6 directs magnetite synthesis and controls size, but until now has not been attempted in flow. Here we report Mms6 increases magnetism, but no difference in particle size is seen, showing flow reduced the influence of Mms6. The study demonstrates a versatile yet simple platform for the synthesis of a vast range of tunable nanoparticles and ideal to study reaction intermediates and additive effects throughout synthesis.

## 1. Introduction

The design and synthesis of precise monodispersed iron oxide magnetic nanoparticles (MNP) is a growing research field due to their applicability in nanotechnologies, particularly in the biomedical sector [1]. In nanomedicine, MNPs comprised of the magnetic iron oxides magnetite (Fe_3_O_4_) or maghemite (γ-Fe_2_O_3_) are considered very attractive due to their low toxicity and cheap precursors. They can act as contrast agents in MRI and can also be heated by an alternating magnetic field or laser light to provide hyperthermia as a remote, switchable therapy for cancer treatment [2,3,4]. MNP can be functionalised with probes (like fluorescent tags for imaging), drugs and targeting or a combination of all of these to form smart theranostic that can be targeted to a specific region of the body by a magnetic field.

However, “green” (ambient, non-toxic conditions) synthesis of iron oxides is notoriously difficult to control. Minor changes to any number of reaction conditions (iron precursors, choice of base, ratio of iron oxidation states of the precursors etc.) will lead to the production of a range of iron oxides [1]. For example, the partial oxidation of ferrous hydroxide using an excess of NaOH under N_2_ at 90 °C will precipitate octahedral magnetite particles approximately 20–80 nm in size. Changing the base used can result in needle-shaped FeOOH by-products [5]. Furthermore, if the excess of NaOH is reduced to stoichiometric base:ferrous ions concentrations the particle size increases dramatically up to approximately 1 µm, as the excess of ferrous ions increases the particle products reduce to 400 nm diameter as the excess of ferrous ions increases [6]. A co-precipitation of ferric and ferrous ions at room temperature under N_2_ with KOH base results in mostly small MNP of poor crystallinity, with a heterogeneous shape and size population ranging from <5 nm up to micrometre scales [2,5]. Although this does allow scope to synthesise various morphologies and sizes, the overwhelming drawback is that it is near impossible to synthesise a mono-dispersed and reproducible product with respect to consistent and monodispersed size and shape. Furthermore, ferric oxide impurities are common (which represent a large proportion of the heterogeneity observed in the co-precipitation synthesis).

A key challenge for the green synthesis of magnetite is the reproducibility of batch nanoparticle synthesis. Very subtle changes in conditions can have huge effects on the final MNP product, which is more exaggerated and difficult to address in batch synthesis, with batch to batch variations being inevitable. 

Micro and macro fluidic systems offer a unilaminar controlled fluid environment, where the fluid flow and dynamics can be quantified, modelled, controlled, and reproduced with higher accuracy, allowing a more reproducible synthesis compared to batch synthesis [7]. Furthermore, defining and achieving precise reaction environments enables a more detailed analysis of the synthesis process, such as the possibility of adding probes for analysis or reagents at precise reactions points. An additional benefit of a continuous fluidic system is that the reaction time points equate to channel position, allowing screening and monitoring of the synthesis at every stage in situ. Such systems can be readily modified to incorporate characterisation instrumentation such as spectroscopy, microscopy and filming to analyse reactions as they occur in real-time at the microscale [8].

Microfluidics have been investigated for the synthesis of inorganic nanoparticles for over a decade [9], but this research has concentrated on the synthesis of quantum dots and noble metal (Ag, Au) nanoparticles [7,8,9,10,11]. To date there have only been two studies that use a fluidics system to fabricate iron-oxide MNP [12,13], which may be surprising considering the obvious benefits controlling the reaction environment has on fastidious iron-oxide synthesis. Abou-Hassen et al. reported some preliminary results for a co-precipitation of magnetite [12]. They reported issues with clogging so utilised a millimetric system, but could not obtain magnetite even when coated with a surfactant to prevent oxidation, instead producing 7 nm sized maghemite coated nanoparticles. In the same year Frenz et al. reported a more sophisticated microfluidic synthesis of iron oxide [13]. Again, the aim was to produce magnetite but they noted it readily oxides to maghemite on contact with air. They utilised aqueous microdroplets of reagents in an organic solvent, where an electric voltage initiates reagent mixing of the droplets to nucleate the formation of 4 nm sized nanoparticles.

While this method is elegant, yields and overall outputs were found to be low. In general, while the small volumes of microfluidic synthesis offer excellent control and reproducibility over batch synthesis, the drawbacks are that these smaller scale syntheses offer smaller quantities of product and issues with clogging of the tubing requires elaborate solutions leading to more complex fabrication of the devices. 

In this work we present a simple (easily castable from PDMS) millimetric macro fluidics device for the synthesis of magnetite MNP’s, and improve on this with a more sophisticated fluidic device machined out of polyether ethyl ketone (PEEK) utilising a glass capillary as the reaction vessel, but of the same simple design (Figure 1a). With the aim of attaining simple green reaction conditions we do not use droplets, organic solvents or surfactants. We explore tuning the particle formation by varying the ratio of ferrous ions to ferric ions in the fluidic synthesis. This demonstrates an excellent simple platform technology for the synthesis of a vast range of nanoparticles with the ability to tune their properties with further development.

Control over the formation of magnetite MNPs have also been achieved in the natural world through biomineralisation using biological additives. In nature many inorganic minerals are formed with exceptional precision over the mineral composition, size and morphology, utilising a range of biomineralisation proteins. One such example is the biomineralisation of magnetite MNP by Magnetotactic bacteria within liposomes in their cells known as magnetosomes [14,15]. Magnetosome membrane specific (Mms) biomineralisation proteins embedded in this liposome are responsible for nucleation of the iron oxide mineral and crystallisation to a precise size and shape [14,15]. Previously, Mms6 has been shown to control the formation of magnetite MNP in vitro when added to green co-precipitation chemical synthesis, showing a promising new methodology of controlling nanoparticle synthesis with biological additives [16,17,18]. It is thought that Mms6 is a nucleating protein for magnetite MNPs [19,20].

In this work we also demonstrate how the fluidics device we produced can be used in conjunction with a biomineralisation protein additive to control the particle formation further. The addition of an Mms protein is a proof-of-concept to demonstrate how such a simple fluidics system could be utilised as a platform to help to understand how proteins (or indeed any other nanomaterial additive) function in vitro in a controlled fluidic environment.

## 2. Materials and Methods

### 2.1. Reagents

Iron (II) sulphate, iron (III) sulphate and sodium hydroxide were purchased from Sigma-Aldrich (Gillingham, United Kingdom) and used without further purification. Iron contents were confirmed via ICP-MS prior to experimentation. PDMS was prepared from Dow Corning (Midland, MI, United States) kit sylgard 184 in the standard ratio 1:10 curing agent:polymer.

### 2.2. Macrofluidics Device Fabrication 

#### 2.2.1. PDMS Device

The device was constructed from a PFTE hollowed out block with a long channel milled from the centre. Initially a stiff metal wire with a diameter of 1.6 mm was pushed through the two holes along the axis of the device. A second input hole, intersected the main channel at an angle of 45° as shown in Figure 1a,c. The blunt end of a needle was shaped using a drill bit to meet and fit the shape of the central wire. Triton X-100 was wiped over the wires/needle so they could be easily removed from the device after casting. The space was then filled with liquid PDMS and cured at 60 °C for 24 h. When cured, the central wire and needle was carefully removed, leaving behind a central channel and side inlet in PDMS. The materials and equipment required to create this device are cheap and readily available, and the PTFE block is reusable for many castings. The block was fastened to a heavy object to prevent lateral movement when connected to stiff fluidic tubes. The fluid from the central channel was fed into a PEEK tube with an inner diameter of 0.02”. The internal hole was expanded using a shaped drill bit to ensure a smooth transition in diameter.

#### 2.2.2. PEEK Device

The device was designed and modelled in SolidWorks, comprising of 6 individual components; (i) Fe inlet, (ii) NaOH inlet, (iii) inlet faceplate, (iv) capillary, (v) outlet faceplate, (vi) outlet (Figure 2).

The rig was machined from PEEK, with o-ring seals between components i–ii, ii–iii, and v–vi to ensure no leakage of solution, a problem often observed in PDMS cast systems. A 27-gauge blunt-end needle (0.41 mm OD, 0.016 ID) was set through component i) with the use of epoxy resin. This needle was sprayed with Teflon spray to aid flow. The capillaries used were 0.5 m in length, with a 5 mm outer diameter and 1.5 mm internal diameter. The capillary tube was inserted between components ii/iii (inlet) and components v/vi (outlet). The system was locked in position with clamp stands to prevent movement when connected to the fluidic tubing.

Fluidic connections for both devices (Upchurch Scientific, purchased from Kinesis (Cambridge, UK) or Fischer Scientific (Loughborough, United Kingdom)) and PEEK tubing with an outer diameter 1/16” and inner diameter of 0.02” were used. Glass capillaries were initially used but were replaced, to reduce problems with clogging, with PEEK capillaries. All tubes were cut with tube cutters (Upchurch Scientific A-327, A-350) to ensure clean and perpendicular cuts. Two syringe drivers (Harvard Apparatus, Cambridge, United Kindrom) were used to control the flow rate of iron and base into the mixer. Glass syringes (SGE Europe Ltd., Milton Keynes, United Kingdom) with volumes of 1 mL and 10 mL were used for the inner and outer flows respectively. Luer fittings were used to connect PEEK tubing to syringes.

### 2.3. Protein Expression and Purification

The *mms6* sequence from Magnetospirillum magneticum AMB-1 was introduced into a pTTQ8 based expression vector by cohesive end cloning with the resulting plasmid, pHis8mms6, encoding N-terminally octahistidine tagged Mms6. The protein was produced in E. coli BL21 star (DE3) cells (Invitrogen, Waltham, MA, United States) harbouring a pRARE (Merck, Nottingham, United Kingdom) plasmid to compensate for codon bias in the mms6 sequence. Cells were cultured in autoinducing Superbroth (Formedium, Hunstanton, United Kingdom) at 37 °C with shaking for 24 h in the presence of carbenicillin and chloramphenicol to select for the pHis8mms6 and pRARE plasmids respectively. Cells were lysed by sonication in 25 mM Tris pH 7.4, 100 mM NaCl. The insoluble material, containing the His8-Mms6 inclusion bodies, was collected by centrifugation at 16,000× *g* and resuspended in 6 M Guanidine Hydrochloride, 25 mM Tris pH 7.4 to solubilise the proteins. Further centrifugation at 16,000× *g* was performed to remove any material not solubilised by the Guanidine treatment. The supernatant was mixed with nickel charged nitrilotriacetic acid (NTA) resin (Amintra resin, Expedeon, Cambridge, United Kingdom) to allow binding of the histidine tagged Mms6. The resin was subsequently packed into a gravity flow column and washed extensively with Wash Buffer (6 M Guanidine hydrochloride, 25 mM Tris pH 7.4, 10 mM Imidazole) before elution of the bound protein in 300 mM Imidazole supplemented Wash Buffer. The eluted protein was refolded by rapidly diluting into a large volume of Refolding Buffer (500 mM NaCl, 25 mM tris pH 7.4) before being concentrated using a 10 kDa molecular weight cut off centrifugal concentrator (Sartorius, Binbrook, United Kingdom). The concentrated material was subjected to centrifugation to remove any small amounts of precipitated protein before dialysis against 500 mM NaCl using a 3.5 kDa molecular weight cut off slide-a-lyser (Thermo Scientific, Waltham, MA, United States). The refolded His8-Mms6 was quantified by absorbance at 280 nm and stored at 193 K.

### 2.4. Continous Flow MNP Synthesis

The device was cleaned with ultrapure water, then dilute hydrochloric acid (1 M) followed by ultrapure water again by pumping 10 mL through both ports.

All reagents were prepared with ultrapure water, and deoxygenated by sparging with N_2_ for 30 min. The outer flow syringe driver was loaded with a 10 mL luer lock syringe of NaOH (1 M) and connected to the co-axial fluidic device via capillary tubing. This was set at a continuous rate of 360 µL/min. The inner flow syringe driver was loaded with a 1 mL luer lock syringe of a mixed ratio of Fe^2+^ and Fe^3+^ salts (ferrous sulphate pentahydrate and ferric sulphate heptahydrate) varied from a 4:1 (0.2 ferric) to 1:2 (0.7 ferric) Fe^2+^:Fe^3+^ ratio with a total iron concentration of (0.05 M) and connected to the co-axial fluidic device via capillary tubing. This was set at a continuous rate of 90 µL/min (although these rates were varied (see results) these were found to be optimum). The solutions were prepared immediately prior to the experiment.

The iron oxide material formed and flowed to the end of the device where it reached the exit port and dripped into a round bottom flask which was kept under an atmosphere of nitrogen. The iron oxide product was magnetically separated and washed 3× in deoxygenated ultrapure water and subsequently dried in a vacuum oven overnight.

Further improvements to the PDMS device included magnetically collecting the iron oxide as it exited the device using a magnetic trap. This was also washed 3× with deoxygenated ultrapure water and dried under vacuum.

### 2.5. Continous Flow MNP Synthesis Modified with Mms6

The iron oxide synthesis was further modified with the addition of Mms proteins, where 50 µg of protein was added to the 1 mL Fe salt solution before the reaction. The experiment then proceeded as before.

### 2.6. Characterisation

#### 2.6.1. Magnetic Susceptibility

Magnetic susceptibility was a measured on a known amount of dry iron oxide nanoparticles using a bench-top Bartington MS2G magnetic susceptometer (Bartington Instruments, Witney, United Kingdom)) at room temperature. The sample was loaded into the instrument in an eppendorf (a blank was subtracted of an empty eppendorf tube). A reading in emu was recorded. Each sample was analysed in triplicate.

#### 2.6.2. Magnetometry

Magnetic susceptibility was performed on a known quantity (1–5 mg) of dry iron oxide nanoparticles on a MPMS 3 SQUID magnetometer (Quantum Design, Surrey, United Kingdom) in vibrating sample mode, with the samples packed in size 3 gelatine capsules. The samples were run at 300 K between −7 and 7 T with a sweep rate of 0.01 T/s. Preliminary magnetic susceptibility measurements were performed on a known quantity (approx. 5 mg) of dry iron oxide nanoparticles using a bench-top Bartington MS2G magnetic susceptometer at room temperature. The sample was loaded into the instrument in an eppendorf (a blank was subtracted of an empty eppendorf tube). A reading in emu was recorded. Each sample was analysed in triplicate.

#### 2.6.3. Transmission Electron Microscopy

10 µL of a 1 mg/mL suspension of nanoparticles in hexane was dropped onto a carbon coating copper TEM grid and allowed to dry down. Grids were imaged using a FEI Tecnai G2 Spirit electron microscope (Thermo Scientific, Waltham, MA, United States) and the TEM images were analysed using Image-J software (v1.52 a, public domain, National Institute of Health, Md, USA). >200 particles per samples were randomly selected and measured. 

#### 2.6.4. X-Ray Diffraction (XRD)

XRD data collected by analysis of dry iron oxide nanoparticles in a Bruker D8 powder diffractometer (Bruker, Coventry, United Kingdom). Diffraction images were collected at 0.022 degree increments from 20–80 degrees, with a fixed wavelength at λ = 1.54178 Å from a Cu Kα X-ray source.

## 3. Results

### 3.1. Design Experimental Set-Up and Optimisation of the Coaxial Flow Device

The co-axial flow device design was based on the work of Abous-Hassen [12], operating under the principle of MNP forming in a sheath flow of sodium hydroxide (NaOH), with a core flow of mixed valence iron salt solution resulting in an axial diffusion gradient between the iron ions and NaOH solution in the centre of the channel. The velocity profile for this coaxial geometry was modelled using the fluid dynamics package in COMSOL Multiphysics. (Appendix A). Recent literature gives further detail to support our modelling, describing how increasing the flow rate of the outer flow, focuses the jet to the centre and thus increases diffusion at the interface [21]. Figure 1b shows the resulting solution to the model showing the cross-sectional velocity in a tube after the junction. It is important to note that the flow regime remains laminar by selecting the correct fluid flow rates. The nanoparticles are formed at the interface which remains stable, no turbulent mixing is required.

The first device was simply cast from polydimethylsiloxane (PDMS) within a Polytetrafluoroethylene (PTFE) holder using a wire and needle to template the channels (Figure 1a,c). In this set up the molten PDMS is simply poured into the PTFE mould and cured over 24 h, then the wire and needle are simply removed. While the PDMS coaxial flow mixer device was based on the example by Hassan et al. [12], a number of design modifications were introduced (see Section 2). The improvements to the construction of the device led to greater ease of use, reliability, and a reduction in the occurrence of blockages and leaking (which are reported across the literature for PDMS microfluidic platforms). Such devices were standardly used for a one to two years timeframe. Over this time, the devices would handle approximately a litre of solution per year. We did not notice any abrasion or cavitation forming in the device over this time and the reproducibility and quality of the data remained consistent over the time. New casting was usually required due to leakage at the junctions. A second generation microfluidic device was produced to the same simple fluidic system design but builds and improves on this by machining the system out of polyether ether ketone (PEEK) to remove the need to re-cast from PDMS which in turn removed the variations between fluidic devices, for greater consistency between reactions (Figure 1d). The junctions in the new device are also designed to reduce/eliminate leakage. Furthermore, by incorporating a glass capillary as the channel through which the reaction occurs, it is possible to vary the retention time of the iron solution in the sheath flow of NaOH solution by altering the capillary length. It is important to note that while the improved machined fluidics device had less issues with clogging and leaking, the profile of the results that follow were reproducible and consistent across both devices (Appendix A).

### 3.2. Optimisation of Coaxial Flow Devices

It is crucial when utilising a sheath flow system for the formation of iron oxide nanoparticles to develop laminar flow of the outer stream (NaOH), allowing it to diffuse into and react with the core stream (iron solution) at the interface. This is achieved by running the NaOH inlet at a faster rate than that of the Fe inlet to ensure an excess of NaOH that can develop into laminar flow before the core stream point of entry. The concentrations of both the NaOH solution and Fe solution can also be varied, meaning the conditions had to be balanced to find both optimum concentrations and flow rates for each of the reagents and inlets.

When the Fe solution had a concentration of 10 mM, an undeveloped inner stream was observed, due to the low amounts of iron. 20 mM, 50 mM, and 100 mM Fe solutions were also tested and found to produce a cohesive flow. 100 mM was selected due to producing the greatest yield of particles in the shortest run time. NaOH concentration was tested at 0.1 M, 0.3 M, 0.5 M, and 1 M with 1 M producing black particles along the length of the reaction capillary.

A minimum Fe/inner flow rate of ~60 µL/min was found to be necessary, as below this rate magnetic particles would begin to form at either the entry to the system in the case of the PDMS moulded device, or at the tip of the needle in the PEEK system, with both cases resulting in clogging and fouling of the system. Incidences of clogging were reduced at an increased inner flow rate of ~90 µL/min. It was found for both devices that when a lower NaOH/outer flow rates was employed (<4:1 ratio between the rates of the two streams), a consistent stream of particles did not develop, with clumps of non-magnetic iron material travelling through the systems instead. A ratio of 4:1 between the outer and inner inlets was found to work optimally (summarised in Appendix A), with black iron oxide particles forming in a stream.

### 3.3. Varying the Ferric:Ferrous Ion Ratio to Tune MNP Magnetism and Size

When synthesising iron oxide MNP’s, the ratio between ferric and ferrous ions have a significant influence on the reaction environment. We have previously found that when the fraction (X) of ferric iron of the total iron (both ferrous and ferric) is varied between X = 0.2–0.7 in a batch synthesis, the iron-oxides produced change, with ferrous rich oxides (such as amorphous ferrous hydroxide and wüstite) being favoured at the lowest ratio, and ferric-rich oxides (such as schwertmannite) being favoured at the higher ratio, with magnetite formation favoured at 0.5–0.6 ferric content [19,22].

In this study we performed the reaction in both fluidic devices varying the fraction of ferric ions from X = 0.2 to 0.7 in line with our previous work [19]. In both cases the 0.6 ratio (3Fe^3+^:2Fe^2+^) gave the highest magnetisation. This was 78.7 emug^−1^, and the 0.5 (1Fe^3+^:1Fe^2+^) ratio producing particles of similar magnetisation at 78.4 emug^−1^ for the PEEK system (see Table 1, Appendix A for comparison with PDMS device), suggesting magnetite was successfully synthesised in the flow system. This is very high for nanoparticles of this size compared with the literature [23], showing the samples to be high quality and purity magnetite. This was confirmed by X-ray diffraction analysis of the resulting particles (Figure 3). The question as to whether or not the particles analysed were formed in the fluidic device or matured in the collection vessel afterwards was addressed by using a magnetic trap to collect the magnetic material straight from the exit of the fluidics channel (see schematic in Appendix A). The nanoparticles from the PDMS device which were initially magnetic and collected in the magnetic trap (primary product) and those collected in the final collection vial (secondary product) were then compared with the total unseparated products. It was clear that there was only negligible magnetism in the secondary product, showing the magnetite nanoparticles were indeed produced in the fluidic system, not matured later in the collection vial (Appendix A). It is interesting to note that the magnetism of the magnetically trapped primary product was virtually the same as the profile for the unseparated product. This means that the non-magnetic iron oxides are also being trapped in such samples, presumably because they are intrinsically aggregated to the magnetite iron-oxides. Importantly, however the samples are not maturing (becoming more magnetic) in the collection vial. The composition of the particles at different ferric/ferric ion ratios was analysed by X-ray diffraction (Figure 3) and compared to the predicted theoretical iron-oxide composition.

The XRD in Figure 3 clearly shows that Magnetite is the main crystalline component for samples at ratio 0.4, 0.5 & 0.6. These data also reveal an absence of contaminating crystalline iron oxides, and this is in agreement with the magnetic data which are all high and similar for these samples. As expected, 0.7 is too high a ratio to produce magnetite as the main product, and as such we observed small, poorly crystalline/amorphous particles. It should be noted that any amorphous products in the sample will not be detectable by XRD. It is impossible to assign a mineral phase to the small, broad, poorly resolved peaks observed, but they could be due to ferric oxides such as schwertmannite, ferrihydrite and feroxyhyte, and to some small quantity of magnetite or maghemite (to account for the small saturation magnetism recorded).

In the more ferrous rich ratios, magnetite is seen alongside ferrous oxides such wüstite and mixed valance oxide green rust. Due to the unknown presence of amorphous oxides ratios of minerals present cannot be extracted from the XRD data, but is can be clearly seen that there is a large proportion of wüstite (and potential green rust) compared to magnetite. Again, the presence of these minerals is responsible for the lower magnetic saturation. The X-ray diffraction pattern could also be used to calculate the average crystalline diameters using the full width half maximum (FWHM) of the 2θ = 35.5° peak data in the Scherrer equation, which are shown in Table 1. Note the 0.7 sample could not be analysed in this way due to the irregularly shaped small peaks. The samples were also analysed by transmission electron microscopy (TEM) and the particles were measured in ImageJ to produce size distributions (Figure 4). We found the average size of the particles formed ranged from 6.5–20.5 nm depending on the initial X-value used, with the lower ratios (0.2–0.3) forming heterogeneous particles with a large variation in sizes observed. Much of this variation is due to the different ferrous-oxide species in these samples. Green rust and ferrous hydroxide forms plates (some of which can be seen in the control X = 0.3 sample in Figure 4). For clarity of sizing, the sizes of these plates were not counted as they would distort the average size of the remaining particles. Wüstite forms irregular shaped particles across a wide size range, which is more difficult reliably recognise to remove from the data, so was not subtracted and so we believe this is responsible for the very large size distribution in these samples. However, there is a visibly bi-modal distribution of smaller and larger particles in the 0.2 and 0.3 samples giving a larger mean size than the higher ratios (Figure 4 and Appendix A) including the magnetite particles. This is not however seen in the XRD sizing. This discrepancy could be due to two factors. The first could be the inclusion of materials that are not magnetite in the TEM sizing as discussed above. The second could be inaccuracies in measuring the XRD peak due to masking. In the XRD the 2θ = 35.5° peak has contributions from both magnetite and wüstite that will serve to artificially broaden the peak, reducing the calculated size of the particles. The particles formed at 0.4–0.6 ratios were found to be the most homogeneous with a reduced standard deviation, and this was expected as these are favourable conditions, with the ratios closest to the natural ratio at which magnetite forms. The magnetic and XRD data show these samples clearly contain majority magnetite and happily also show close agreement between the XRD sizing and the TEM sizing (Table 1), showing the crystallite size is the same as the particle size, indicating single crystalline particles. What is most interesting is that particle size reduces as the X value increases. This is clearly seen in both TEM and XRD sizing for this magnetite dominated samples. This shows that (with careful optimisation) the size of magnetite nanoparticles could be subtlety tuned by varied starting ratio of ferric to ferrous iron precursors within this 0.4–0.6 range. It should be noted that particle size also affected the saturation magnetisation. However, recent studies show small size difference between 13 and 9 nm will have little effect in the superparamagnetic regime [23]. Furthermore, the trend is the opposite of what we observe in our data. Identical samples show increased magnetic saturation with increasing particle size, whereas our samples show increasing magnetic saturation as size decreases as ratio goes from 0.4–0.6. As such, the difference in magnetic saturation seen can only be due to increased quality and quantities of magnetite nanoparticles present. Particles formed at the 0.7 ratio were the smallest 6.5 ± 3.0nm, however, the significant reduction in magnetism of this sample suggests the majority is not magnetite but non-magnetic iron oxides, so while it still follows the trend, the size of this sample is not as relevant when considering tuning the size of magnetite particles.

### 3.4. Addition of Mms6 

Mms6 has previously been found to influence the phase and size of particles formed in a batch co-precipitation of magnetite. Extensive research of this protein has shown it to self-assemble into micelles in solution, displaying an acidic C-terminal surface to nucleate the forming iron oxides. We thus investigated if Mms6 would have the same controlling effect in a fluidic synthesis. Again, the synthesis was conducted across the 0.2–0.7 X ratio to draw a comparison between control particles and those formed in the presence of Mms6 protein. In this synthesis 50 µg of Mms6 was added to the 50 mM iron solution (1 mL) before flowing through the device. Very little difference was seen in morphology between those particles formed with and without Mms6 (Figure 4). This was also true of the size (Figure 5a, see Appendix A for detailed particle size data) showing no notable difference, except with the 0.2 samples. Here we see more control towards smaller particles more similar to higher ratio samples. However, more difference is observed in the magnetic properties of the particles formed (Figure 5b). Interestingly, particles formed in the presence of Mms6 showed increased magnetic saturation for all initial iron ratios than the control, except for the 0.6 ratio. As there is no difference in size, this difference is solely due to the improved quality in Mms6 mediated samples from 02–0.5 over controls for additional/larger TEM images of all samples see Appendix A. This is in complete agreement with our previous studies in a batch synthesis [19]. XRD analysis of these samples shows that there is indeed more magnetite present in the most extreme ratios of 0.2 and 0.7 (Figure 6).

The X-ray diffraction data correlates well with the magnetic data, showing a now visible peak at the dominant magnetite 311 reflection of 2θ = 35.5° in the Mms6 0.7 ratio sample along with close to doubled increase in saturation magnetisation (from 10 to 34 emu/g). Furthermore, the magnetite peaks are clearly larger than the wüstite peak in the 0.2 Mms6 sample compared to the 0.2 control sample. This again is matched by an increase in saturated magnetic moment from 30.9 to 41 emu/g, suggesting that Mms6 is aiding the formation of magnetite at unfavourable X ratios, with a similar trend observed to our previous studies [19].

## 4. Discussion

A key issue that occurs in many microfluidic systems is clogging and fouling, and in the case of PDMS systems, leakage. The problem of clogging is exacerbated by the magnetic nature of the formed iron oxide products, resulting in magnetic aggregation which leads to clogging and obstruction of laminar flow. These issues were addressed in PDMS systems by increasing the diameter of tubing, ensuring the fluidic channel is as straight as possible, and by shaping the needle used to cast the device for a smoother co-axial junction. However, this system was still prone to leaking so the design of a second PEEK system addressed this issue by utilising a capillary as the reaction vessel joined to the inlets by specifically machined connectors, sealed with o-rings. The transparent nature of the material of the PEEK system also allows the reaction lifetime to be potentially monitored with optical microscopic or spectroscopic techniques with greater ease. Both devices showed consistent results from run to run within the same device and also across the two devices (once the flow rates were optimised) (see Appendix A), showing the formation of magnetite in this simple fluidic system has good reaction control.

Magnetite is stoichiometrically formed of 2 Fe^3+^:1 Fe^2+^ giving an X ratio of 0.667. Oxidation is known to occur when performing this reaction so initial precursor ratios of 0.5 and 0.6 are commonly used. It is clear from this study that X = 0.4–0.6 give a good yield of magnetite, demonstrated by the magnetism, TEM and XRD, with lower X-values contaminated with ferrous iron oxides such as wüstite and green rust (0.2 and 0.3). It is also worth acknowledging that XRD only detects crystalline materials so amorphous phases could also be present. There has been detailed theoretic and experimental research into the iron oxides that form at different ferrous:ferric precursor ratios [19,22]. The results in this paper tally well with previous findings where mixtures of ferrous hydroxides, ferrous oxides, green rust and a small amount of magnetite can be formed at lower ratios while mixtures of magnetite and ferric-oxides such as schwertmannite and goethite are seen in higher ratios (above the stoichiometric ratio). It is excellent to see the fluidic system reproduces the results from previous batch studies and tally well with theoretical work. However, what has not been seen before is the ability to tune the size with varying X. It is not clear from searching the literature if this has not been explored before in all synthetic forms of precipitating magnetite, batch, fluidic or otherwise, or whether this effect is something only seen in a flow synthesis. In the flow the iron solution is been fed the base at the interface at a continuous rate at the interface were the crystals will be seeded. Perhaps at ratios closer to stoichiometric, the nucleation is so rapid that the particles do not have the capacity to grow quite as much before all the iron solution is used up, forming smaller particles. However, in the lower ratios, fewer particles can be nucleated giving more iron feedstock to grow the smaller number of particles more.

We developed the experiment further by adding the biomineralisation protein Mms6 to the flow synthesis. In agreement with our previous study this aided the formation of magnetite at ratios where the control showed less magnetite [19]. This is thought to be possible as the acidic rich DEEVE amino acid motif on the C-terminus of the protein is exposed and is capable of binding iron ions [19,20,24,25]. The N-terminus is hydrophobic and as such causes the protein to self-assemble into rafts within the native environment or micelles in aqueous solution, displaying the acidic residues as a negative charged nucleation surface for iron binding. Importantly it has been show to bind ferrous ions in some sites and ferric ions less specifically but in abundance, encouraging the nucleation of magnetite specifically [20]. Previous work has also shown that Mms6 controls the size of the particles to be 20 nm when formed in solution. However, we did not see this control in the flow synthesis. We hypothesis that the synthesis in flow increases the effect of the chemical kinetics of particle formation over slower additive modified crystal growth giving little room for Mms6 to be effective, especially with respect to particle size control. However interestingly, it has been previously reported that the curvature of the charged surface may be key to size controlling aspect of Mms6. When Mms6 is displayed on a flat surface the particles formed are larger (90 nm) compared to 20 nm in solution, so it could also be the case that Mms6 in the flow system cannot form the correct size-controlling surface [26].

Finally, this device, experiments and results demonstrates how this system could be a beneficial platform for the study of nanoparticle formation more generally. The reaction vessel is long so time points can be equated to distance. This can be useful in both experiment design (introducing additive such as Mms6 at different reaction points) and analysis (studying the reaction intermediates in situ). Similarly, this could be universally extended to study the action and effect of a full range of different additive and at different stages in a wet chemistry synthesis of any nanoparticles.

## 5. Conclusions

We have successfully built a simple and reliable fluidic system that reproducibly formed magnetite nanoparticles. Magnetite is the major product in the synthesis where the ferric to total iron ratio of the precursor solution was 0.4–0.6, demonstrated by both the XRD and the magnetic data. The size of the nanoparticles can be tuned simply by varying this ratio, however consistent magnetite and thus tuning is best achieved between ferric fractions of 0.4–0.6 (with larger particles at higher ferrous content (X = 0.4) and smaller particles with higher ferric content (X = 0.6). Outside this range other iron oxides were obtained, reducing the homogenity in terms of size, and morphology, as well as reducing the magnetic saturation. The more ferrous rich ratios (X = 0.2–0.3) contained wüstite and green rust, while the phases of the more ferric rich X = 0.7 ratio were unidentifiable from our characterisation. Addition of Mms6, a magnetite nucleation protein, helped to produce more magnetite within all samples (except X = 0.6) with the most profound effect at the 0.2 and 0.7 extremes. Mms6 was not able to affect the particle size in the fluidic system, which it is able to do in a batch synthesis. We speculate this may be due to the complex competing effects of the dominance of kinetics on synthesis in dynamic flow, coupled with the flow conditions affecting the Mms6 assembly. This study shows a promising proof-of-concept for using a simple fluidic system for the formation of and detailed in situ analysis of the formation of a full range of inorganic nanomaterials, and their interaction with crystallization additives, such as Mms6.

## Figures and Tables

**Figure 1 nanomaterials-09-01729-f001:**
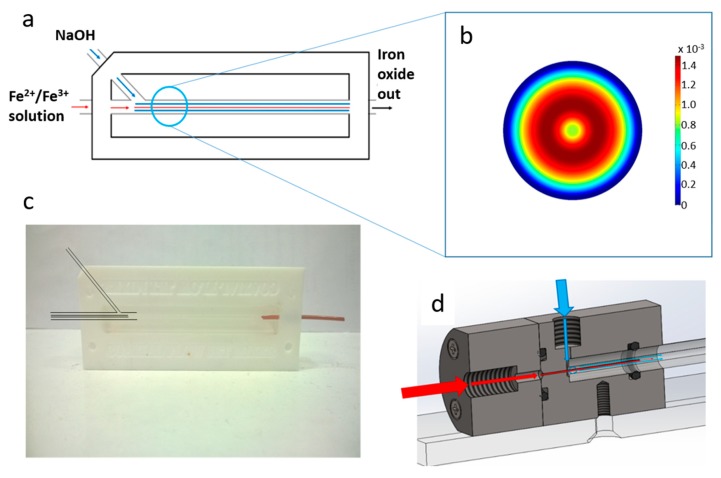
(**a**) schematic of the design of the co-axial fluidic device, showing the fluidic junction between the two streams of reactants. (**b**) Cross-sectional velocity of sheath flow around a core flow in a tube after the junction, (**c**) Photograph of the polydimethylsiloxane (PDMS) device with inlets highlighted for clarity, (**d**) cross-sectional view of polyether ethyl ketone (PEEK) device. For (**a**,**d**) iron/NaOH inlet with red arrows representing the flow of iron solution and blue arrows representing the flow of NaOH solution.

**Figure 2 nanomaterials-09-01729-f002:**
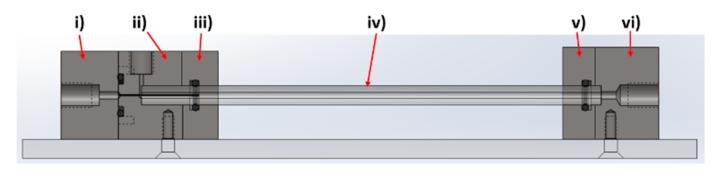
Cross section of PEEK system illustrating the different components.

**Figure 3 nanomaterials-09-01729-f003:**
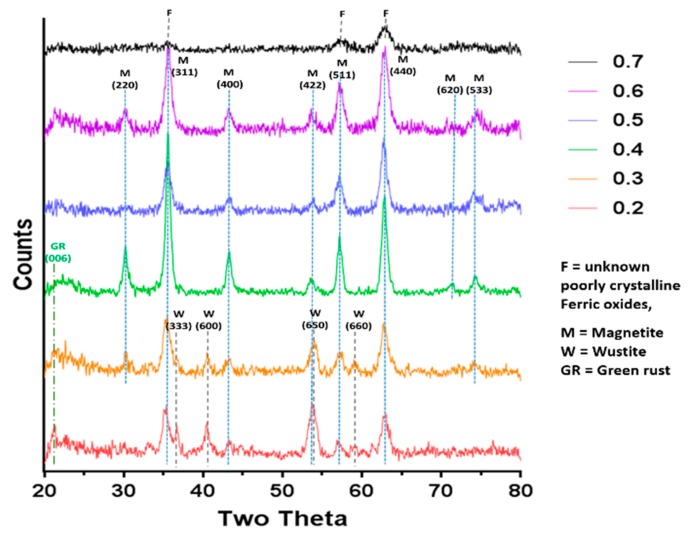
Annotated X-Ray Diffraction (XRD) data of the control samples across the range of ferric to ferrous ratio.

**Figure 4 nanomaterials-09-01729-f004:**
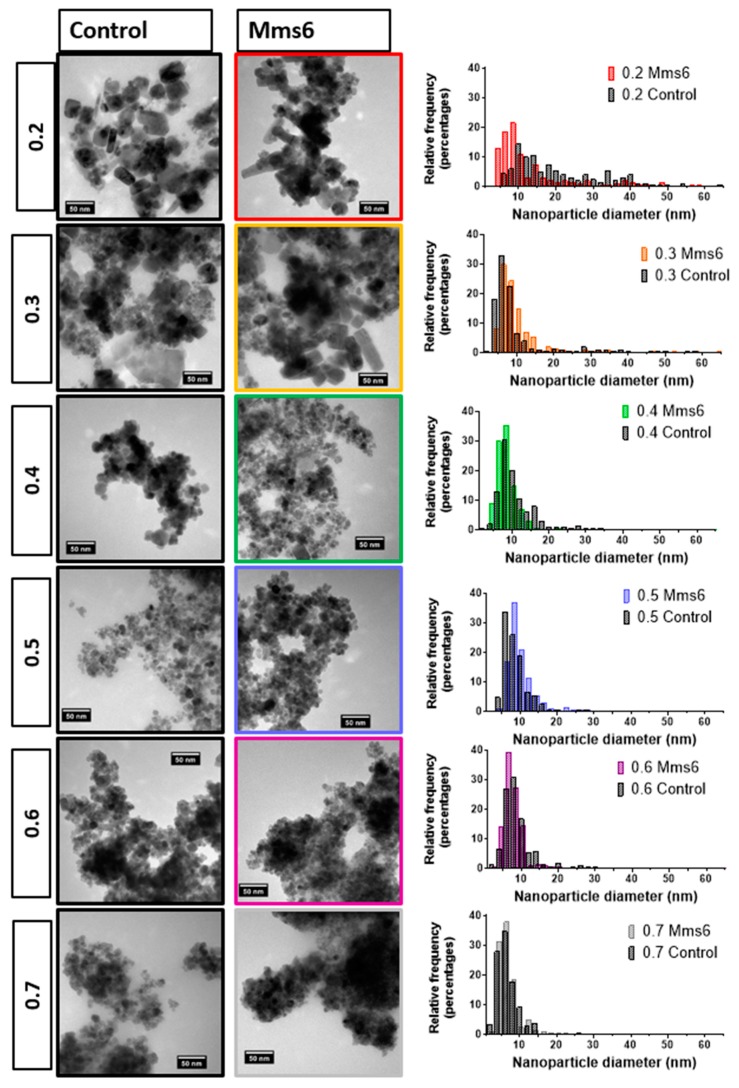
Representative transmission electron microscopy (TEM) image of the particles formed in the fluid device over the 0.2–0.7 ratio of ferric iron:total iron. Left column control samples and right column with the addition of 50 µg Mms6.

**Figure 5 nanomaterials-09-01729-f005:**
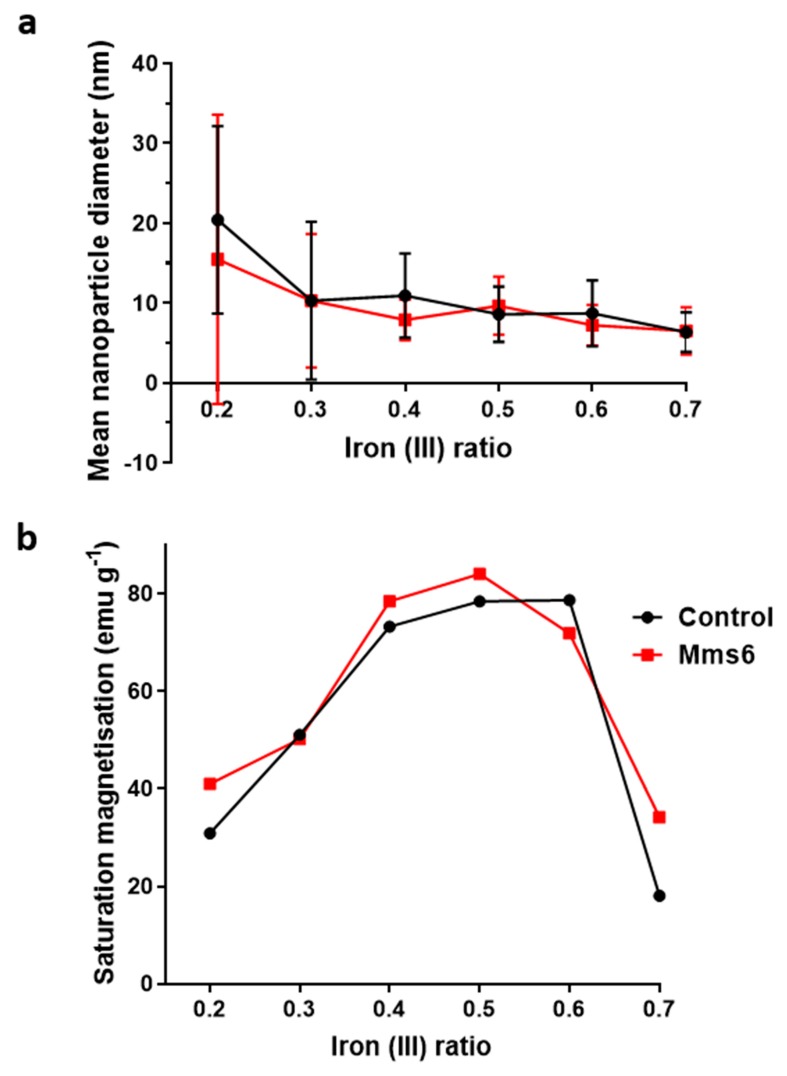
Comparison of control particles to those formed in the presence of 50 µg of Mms6 within the fluidic system. (**a**) Variation of particle size with increasing initial ratio of ferric ions. (**b**) Variation of saturated magnetic moment with increasing initial ratio of ferric ions.

**Figure 6 nanomaterials-09-01729-f006:**
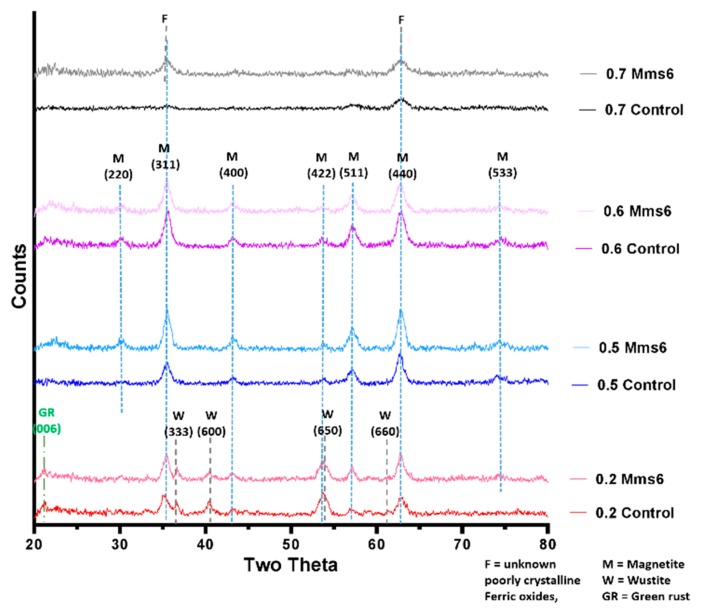
XRD of particles formed in control and Mms6 reactions for 0.2, 0.5, 0.6 and 0.7 X ratio reactions, annotated to show iron oxide.

**Table 1 nanomaterials-09-01729-t001:** Tables of characterization of all the control samples from 0.2–0.7 ferric ion fraction to total ferric and ferrous ion.

Sample	Major Crystalline Iron Species	Particle Size XRD (nm)	Particle Size TEM (nm)	Sat. Magnetic Moment (emu/g)
0.2	Magnetite, Wüstile, Green rust	9.32	13.1 ± 9.6 (35.7 ± 19.4) Mean 20.5 ± 11.8	30.89
0.3	Magnetite, Wüstile, Green rust	9.32	6.9 ± 4.9 (31.2 ± 10.2) Mean 10.9 ± 10.3	51.06
0.4	Magnetite, Green rust	13.31	10.9 ± 5.3	73.27
0.5	Magnetite	10.36	8.6 ± 3.5	78.44
0.6	Magnetite	9.32	7.4 ± 4.1	78.67
0.7	Poorly crystalline ferric oxides	N/A	6.5 ± 3.0	18.08

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
