# Peer review of "Macrofluidic Coaxial Flow Platforms to Produce Tunable Magnetite Nanoparticles: A Study of the Effect of Reaction Conditions and Biomineralisation Protein Mms6"

_nanomaterials, 2019, doi:10.3390/nano9121729_

Round 1

Reviewer 1 Report

The article „Macrofluidic Platforms to Produce Tunable Nanoparticles: Coaxial flow synthesis of magnetite nanoparticles and study of particle size as a function of reaction conditions and biomineralisation protein Mms6 additive.“ from Laura Norfolk, Andrea Rawlings, Jonathan Bramble, Katy Ward, Noel Francis, Rachel Waller, Ashley Bailey, Sarah S. Staniland deals with the microfluidic synthesis of magnetite nanoparticles. Besides the design of the developed platform for synthesis, the influence of different parameters on the particle size has been described. Overall, the paper is understandable written and well structured. The paper is addressed for experienced readers as well as newcomers in the field of flow synthesis of magnetite nanoparticles. To improve the manuscript, the reviewer suggests the following points: Line 2-7: shorten the title! Line 18: caution: the flow in macrofluidic systems is rarely laminar, please specify Line 28: “unlimited range of tunable nanoparticles” sounds a bit too optimistic referring to the achieved changes of particle size ranges Line 62-64: yes, but less output! Figure 1: a) a better quality of the image is desirable Line 306: remove “should be placed in the main text…” Line 346: concerning “indicating highly crystalline particles”: you have done TEM characterization - why there is no high-resolution TEM or diffraction image of the particles given? There you can check immediately the crystallinity. Line 391: The saturation magnetization is for small particles (< 10 nm) a function of particle size. Please consider this point. A microfluidic simulation of the fluid flow in the platform is missing in this manuscript, because the authors are claiming the existence of a laminar flow in their structure and at the same time, they are talking about the mixture of their reagents. It is very tricky to mix fluids in laminar conditions, so please clarify this point - otherwise it sounds like a contradiction in terms.

Author Response

Response to Reviewer ~#1

I would like to thank reviewer 1 for the time you have dedicated to our manuscript. We are delighted to see that reviewer 1 has answered a positive “yes” to the 5 key questions and favourable comments “Overall, the paper is understandable written and well structured. The paper is addressed for experienced readers as well as newcomers in the field of flow synthesis of magnetite nanoparticles”. They have raised some points to address, which I will address in turn below: 

Line 2-7: shorten the title!

This is a fair point and we have now shortened the title considerable to “Macrofluidic Coaxial Flow Platforms to Produce Tunable Magnetite Nanoparticles: A Study of the Effect of Reaction Conditions and Biomineralisation Protein Mms6.”

Line 18: caution: the flow in macrofluidic systems is rarely laminar,

See your last point below. We have added clarity with the text: “formation at the interface by diffusion between two laminar flows,”

Line 28: “unlimited range of tunable nanoparticles” sounds a bit too optimistic

We concede unlimited is a somewhat oversell. We do below this fluidic synthesis  could have far reaching implications on many different types of nanoparticle synthesis, so have replaced “unlimited range” with “a vast range”

Line 62-64: yes, but less output!

Yes this is true, so we have added “at the cost of reduced output” into the text

Figure 1: a) a better quality of the image is desirable

It is fair to say the text on figure 1a is too small to be legible. I think this is also true for the scale in figure 2 B. We have thus increased the size and quality of this to improve figure 1.

Line 306: remove “should be placed in the main text…”

We are embarrassed this note was spotted and removed by us before submission. This has now been removed.

Line 346: concerning “indicating highly crystalline particles”: you have done TEM characterization - why there is no high-resolution TEM or diffraction image of the particles given? There you can check immediately the crystallinity.

Perhaps the phasing of this sentence is misleading. We were drawing the conclusion that if the crystalline size from the XRD is the same as the size from the TEM then the particels themselves are not made up of several crystallites but are indeed single crystalline particles. We have now changed the wording to say this explicitly. Now text now reads: “show close agreement between the XRD sizing and the TEM sizing (Table 1), showing the crystallite size is the same as the particle size, indicating single crystalline particles.” HRTEM could be collected but this this very time consuming and would delay the publication. I also have reservations as I don’t think it would add anything extra, as a diffraction pattern from one particle is not as encompassing as the XRD data for the whole sample. The decision for if this is required rests with the editor.

Line 391: The saturation magnetization is for small particles (< 10 nm) a function of particle size. Please consider this point.

The saturation magnetization is a function of particle size for all szies of nanoparticles not just those less than 10 nm. In fact in the superparamagnetic regime, the effect of size is much (see paper Li, Q.; Kartikowati, C.W.; Horie, S.; Ogi, T.; Iwaki, T.; Okuyama, K. Correlation between particle size/domain structure and magnetic properties of highly crystalline Fe3O4 nanoparticles. Scientific Reports 2017, 7, 9894, doi:10.1038/s41598-017-09897-5). Interestingly though our data shows increased Ms as particle size reduces, which is the opposite to what should happen for identical samples of decreasing size. This further demonstrates the increasing Ms is due to larger quanities/ better quality magnetite in samples 0.5-0.6 as it overcomes the issue of size. I however below the size issues is negligible in our range.  However, it is a valid point to consider and we have thus added the following text into the paper:

“It should be noted that particle size also affected the saturation magnetisation. However, recent studies show small size difference between 13 and 9 nm will have little effect in the superparamagnetic regime.[23] Furthermore, the trend is the opposite of what we observe in our data. Identical samples show increased magnetic saturation with increasing particle size, whereas our samples show increasing magnetic saturation as size decreases as ratio goes from 0.4-0.6. As such, the difference in magnetic saturation seen can only be due to increased quality and quantities of magnetite nanoparticles present.”

It is thus also note worth that any difference seen in the Ms of control and Mms6 particles is again due to improvement in the material and not size as there is no significant difference in size. This has also been highlighted with additional text

“As there is no difference in size, this difference is solely due to the improved quality in Mms6 mediated samples from 02-0.5 over controls.”

A microfluidic simulation of the fluid flow in the platform is missing in this manuscript, because the authors are claiming the existence of a laminar flow in their structure and at the same time, they are talking about the mixture of their reagents. It is very tricky to mix fluids in laminar conditions, so please clarify this point - otherwise it sounds like a contradiction in terms.

The reviewer is correct that we need to clarify our writing on this point to make it clear to the reader. We have modelled the system in COMSOL in order to design the flow regime we use (faster flow rate on the outer NaOH stream) and our modelling as well as established literature (Andreev, V.P.; Koleshko, S.B.; Holman, D.A.; Scampavia, L.D.; Christian, G.D. Hydrodynamics and Mass Transfer of the Coaxial Jet Mixer in Flow Injection Analysis. Analytical Chemistry 1999, 71, 2199-2204, doi:10.1021/ac981037t.) show the flows to be laminar and it is the diffusion between these two laminar flows where mixing occurs. We appreciate this many not have been clear in the original text so have added/modified the following text, and on the suggestion  of the reviewer added the mesh and COMSOL modelling into the supplementary data.

“The co-axial flow device design was based on the work of Abous-Hassen[12], operating under the principle of MNP forming in a sheath flow of sodium hydroxide (NaOH) and a core flow of mixed valence iron salt solution, resulting in an axial diffusion gradient between the of iron ions and the NaOH solution in the centre of the channel. The velocity profile for this coaxial geometry was modelled using the fluid dynamics package in COMSOL Multiphysics. (Supplementary figure S1). Recent literature gives further detail to support our modelling, describing how increasing the flow rate of the outer flow, focuses the jet to the centre and thus increases diffusion at the interface [21]. Figure 1b shows the resulting solution to the model showing the cross-sectional velocity in a tube after the junction. It is important to note that the flow regime remains laminar by selecting the correct fluid flow rates. The nanoparticles are formed at the interface which remains stable, no turbulent mixing is required.”

We believe we have addressed all the reviewer’s comments and incorporated the necessary changes, required. We think the manuscript is now ready for publication.

Reviewer 2 Report

The manuscript is well written and topical. It is bringing valuable novelty.

I recommend accepting after minor revision.

Typos:

1, Regarding the PDMS part, could authors mention what is the volume of the working solution going through before any marks of the abrasion or cavitation occurs? PDMS is much softer material than PEEK.

2, The authors mentioned the possible use of MNPs for diagnostics (theranostics) which is ok for small diameter size around 5nm meeting the condition that conjugated molecule (e.g. labeling molecules) is also low-molecular, all due to possible penetration through the cell membranes in the body. But in the other way there is mentioned (row 39) that also targeting agents as antibodies are transportable in vivo. I´am afraid that Ab has more than 10nm size (depending on the type), and while connected to NPs via linkers it can easily get more than 15 to 30 nm (when counting the spherical shape of conjugate and maximizing the surface coverage), which is hardly acceptable to be transportable as a drug in the body. Could author add an information how is such usage possible (e.g. using of transporters as liposomes, transporting proteins, or different one which are suitable for human body?).

96..."(c)Photograph" (gap is missing)

454..."homogeneity" (homogenity)

Author Response

Response to Reviewer ~#2

I would like to thank reviewer #2 for the time you have dedicated to our manuscript. We are delighted to see that you believe “The manuscript is well written and topical. It is bringing valuable novelty. I recommend accepting after minor revision”. 

The reviewer mentions 2 areas for revision:

The reviewer asked us to “mention what is the volume of the working solution going through before any marks of the abrasion or cavitation occurs? PDMS is much softer material than PEEK.”

The reviewer is of course correct, PDMS is indeed much softer than PEEK. However, we did not witness any marks or abrasion on our devices. We tended to use each devices for about 1-1.5 years before casting a new one. This was mainly because of leaking at the fittings. We did not notice a reduction in quality of data or reproducibility in early data compared to data collected after approximately I year. We do not have the old devices now to assess them but have added the following text to the paper to address this request.

“Such devices were standardly used for a one to two years timeframe. Over this time, the devices would handle approximately a litre of solution per year under low flow rates. We did not notice any abrasion or cavitation forming in the device over this time and the reproducibility and quality of the data remained consistent over the time. New casting was usually required due to leakage at the junctions.”

The authors mentioned the possible use of MNPs for diagnostics (theranostics) which is ok for small diameter size around 5nm meeting the condition that conjugated molecule (e.g. labeling molecules) is also low-molecular, all due to possible penetration through the cell membranes in the body. But in the other way there is mentioned (row 39) that also targeting agents as antibodies are transportable in vivo. I´am afraid that Ab has more than 10nm size (depending on the type), and while connected to NPs via linkers it can easily get more than 15 to 30 nm (when counting the spherical shape of conjugate and maximizing the surface coverage), which is hardly acceptable to be transportable as a drug in the body. Could author add an information how is such usage possible (e.g. using of transporters as liposomes, transporting proteins, or different one which are suitable for human body?).

The example of attachment to antibodies has been removed to avoid any confusion.

We are very grateful for the considered proof-reading the reviewers have done to notice additional typographical errors and we are very pleased to amend these. All additions are amended as tracked changes in the manuscript.

Typos

96..."(c)Photograph" (gap is missing) :- a space has been added

454..."homogeneity" (homogenity): is now corrected spelt

We believe we have addressed all the reviewer’s comments and incorporated the necessary changes, required. We think the manuscript is now ready for publication.

Round 2

Reviewer 1 Report

The manuscript "Macrofluidic Coaxial Flow Platforms to Produce Tunable Magnetite Nanoparticles: A Study of the Effect of Reaction Conditions and Biomineralisation Protein Mms6" has been revised thoroughly and is now ready for publication.

Reviewer 2 Report

The MS entitled "Macrofluidic Platforms to Produce Tunable Nanoparticles: Coaxial flow synthesis of magnetite nanoparticles and study of particle size as a function of reaction conditions and biomineralisation protein Mms6 additive." by Laura Norfolk et al, is well written and after succesfully passed minor revisions I recommend it for publication in Nanomaterials, where it certainly contribute as a novel methodical aproach in to the are of production of magnetic spherical nanoparticles.